# Distillation of a tractable model from the VQ-VAE

**Armin Hadžić** [1]     **Milan Papež**[1]     **Tomáš Pevný**[1]

[1]Artificial Intelligence Center, Czech Technical University. Prague, Czech Republic.

## Abstract

Deep generative models with discrete latent space, such as the Vector-Quantized Variational Autoencoder (VQ-VAE), offer excellent data generation capabilities, but, due to the large size of their latent space, their probabilistic inference is deemed intractable. We demonstrate that the VQ-VAE can be *distilled* into a tractable model by selecting a subset of latent variables with high probabilities. This simple strategy is particularly efficient, especially if the VQ-VAE underutilizes its latent space, which is, indeed, very often the case. We frame the distilled model as a probabilistic circuit, and show that it preserves expressiveness of the VQ-VAE while providing tractable probabilistic inference. Experiments illustrate competitive performance in density estimation and conditional generation tasks, challenging the view of the VQ-VAE as an inherently intractable model.

## 1 INTRODUCTION

Deep generative models provide a framework for learning complex patterns across diverse data modalities, such as images, language, and graphs [Alaniz et al., 2022, Tian et al., 2024, Bachmann et al., 2024]. They excel at generating new samples; however, they cannot answer even the most basic inference tasks without approximations, making them *intractable probabilistic models*. This issue arises because deep neural networks make analytical integration, which is crucial for many inference tasks, infeasible [Nguyen and Goulet, 2021, Rezende et al., 2014]. Probabilistic circuits (PCs)[Choi et al., 2020, Peharz et al., 2020, Vergari et al., 2015] are *tractable probabilistic models* that provide closed-form solutions to a wide range of inference tasks, including marginalization, conditioning, and expectation. Recent work has begun exploring hybrid frameworks that combine the

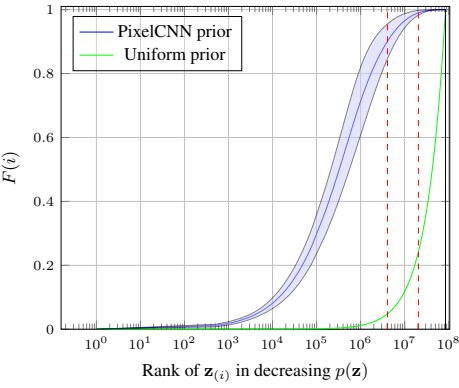

Figure 1: *Cumulative distribution functions (CDFs) of different prior distributions, $p(\mathbf{z})$, in the VQ-VAE.* The CDF is defined as $F(i) = \sum_{j=1}^{i} p(\mathbf{z}_{(j)})$, where $\mathbf{z}_{(j)}$ is the $j$ th latent variable in a decreasing order which is obtained based on $p(\mathbf{z})$ values. The VQ-VAE's latent space size is $|\mathcal{Z}| = 96^4 \approx 85\text{M}$. We repeated the experiment with five independently trained models. The solid line is the mean and the shaded area is the $\pm 1$ standard deviation. The blue and green colors correspond to the PixelCNN and the uniform prior, respectively. The red vertical lines are sizes of the latent space at which $F$ reaches $90\%$ and $99\%$ of its total mass. Specifically, these lines correspond to approximately $5\%$ and $25\%$ of the latent space. This result shows that only a small fraction of the vast VQ-VAE's latent space is actually utilized.

efficient inference of PCs with the expressive power of deep neural networks [Correia et al., 2023, Liu et al., 2023, Sidheekh et al., 2023]. This paper extends this line of work by investigating the tractability of the vector-quantized variational autoencoder (VQ-VAE).

The VQ-VAE [van den Oord et al., 2017] is a deep generative model that compresses input data into a discrete latent representation, while preserving structural patterns (e.g., spatial coherence in images [van den Oord et al., 2017], temporal consistency in audio and video [Dhariwal et al., 2020, Zeghidour et al., 2021, Yan et al., 2021]). However, the latent space of the VQ-VAE is exponentially large, making probabilistic inference tasks intractable [Correia et al., 2020, Zhao et al., 2016, Peharz et al., 2016]. Moreover, the vanilla

*Accepted for the 8th Workshop on Tractable Probabilistic Modeling at UAI (TPM 2025).*

VQ-VAE is known to suffer from the index collapse issue, which manifests as a poor utilization of the latent space. This underutilization means that only a (relatively) small collection of latent states contributes to the model outputs (e.g., its likelihood) [Guo et al., 2024, Huh et al., 2023]. Various techniques have been explored to mitigate this problem, namely replacement policies [Zeghidour et al., 2021, Dhariwal et al., 2020], codebook resets [acucki et al., 2020, Zeghidour et al., 2021, Dhariwal et al., 2020], stochastically-quantized VAE [Takida et al., 2022, Roy et al., 2018], exponential moving average [van den Oord et al., 2017], and product-quantized VAE [Guo et al., 2024].

Notwithstanding that, we advocate that the latent space underutilization, caused by the index collapse, is actually beneficial for the tractability of the VQ-VAE, i.e., the limited use of the latent space makes many inference tasks tractable. We propose a novel approach to transform the VQ-VAE into a tractable mixture model (i.e., a PC) by distilling a subset of the most relevant latent variables. The subset is identified using two complementary approaches: (i) random sampling, which requires enumeration of all latent variables, which is exhaustive and computationally expensive; and (ii) a beam search, which trades off optimality for computational efficiency. We demonstrate that the distilled model based on the beam search delivers a competitive performance to other tractable probabilistic models in the context of learning a probability distribution of images.

## 2 BACKGROUND

**Discrete latent variable models.** A complete probabilistic description of a discrete latent variable model is given by the following marginal probability distribution:

$$p(\mathbf{x}) = \sum_{\mathbf{z} \in \mathcal{Z}} p(\mathbf{x}|\mathbf{z})p(\mathbf{z}), \qquad (1)$$

where $\mathbf{x} := \{x_1, \ldots, x_S\} \in \mathcal{X}$ denotes observations with $S$ elements (e.g., image pixels), and $\mathbf{z} := \{z_1, \ldots, z_M\} \in \mathcal{Z}$ are discrete latent variables. The prior $p(\mathbf{z})$ is a probability mass function over discrete latent variables, describing the significance of each value of $\mathbf{z}$. The conditional distribution $p(\mathbf{x}|\mathbf{z})$ models $\mathbf{x}$ conditionally on a different set of parameters depending on $\mathbf{z}$, i.e., $p(\mathbf{x}|\mathbf{z}) := p(\mathbf{x}|\theta_{\mathbf{z}})$. The latent space, i.e., the set of all latent variable configurations, is a finite structured set with exponential complexity [Correia et al., 2020, Zhao et al., 2016, Peharz et al., 2016].

**Probabilistic circuits.** A *probabilistic circuit* is a directed, acyclic, parametrized computational graph encoding a non-negative function over observations, $p(\mathbf{x})$ [Vergari et al., 2019, Choi et al., 2020]. The graph composes of three types of computational *units*: *input*, *product*, and *sum*. Sum and product units receive the outputs of other units as inputs. We denote the set of inputs of a unit $u$ as $\text{in}(u)$. Each unit $u$ encodes a function $p_u$ over a subset of random variables $\mathbf{x}_u \subseteq \mathbf{x}$, referred to as *scope*. The input unit $p_u(\mathbf{x}_u)$ computes a pre-defined, parametrized probability distribution. The sum unit computes the weighted sum of its inputs $p_u(\mathbf{x}_u) := \sum_{i \in \text{in}(u)} w_i p_i(\mathbf{x}_i)$, where $w_i \in \mathbb{R}$ are the weight parameters, and the product unit computes the product of its inputs, $p_u(\mathbf{x}_u) := \prod_{i \in \text{in}(n)} p_i(\mathbf{x}_i)$. The scope of any sum or product unit is the union of its input scopes, $\mathbf{x}_u = \bigcup_{i \in \text{in}(u)} \mathbf{x}_i$ [Vergari et al., 2019, Choi et al., 2020]. To achieve tractable inference, the children of each sum unit have to be defined over the same scopes (smoothness), and the children of each product unit have to be defined over pairwise disjoint scopes (decomposability). Additionally, the input units have to be tractable probability distributions (e.g., a member of an exponential family) [Darwiche and Marquis, 2002]. PCs are an example of the discrete latent variable model in (1), where the sum units parametrize $p(\mathbf{z})$, and the input units define $p(\mathbf{x}|\mathbf{z})$. Importantly, the decomposability of the product unit imposes conditional independence in $p(\mathbf{x}|\mathbf{z})$, which is the key to tractable analytical integration of arbitrary subsets of $\mathbf{x}$ in (1).

**VQ-VAE.** The vector quantized-variational autoencoder (VQ-VAE) [van den Oord et al., 2017] is a deep probabilistic model for discrete representation learning. The model is similar to the variational autoencoder [Kingma and Welling, 2022] but differs primarily by its vector quantization block. This block uses a collection of latent embedding vectors, $\{\mathbf{e}_i\}_{i=1}^K \subset \mathbb{R}^D$, and is referred to as the *codebook*, where $D$ is the codeword length, and $K$ is the codebook size. The encoder network $\text{E} : \mathbb{R}^{d \times h \times w} \to \mathbb{R}^{D \times H \times W}$, compresses $\mathbf{x}$ into a continuous latent variable, where $M = HW$ are dimensions of the latent space. After the compression, the continuous latent variable is mapped to a discrete latent variable, $\mathbf{z} \in \mathcal{Z}$, by a nearestneighbor search in the codebook using Euclidean distance.

**Definition 1.** *The discrete latent space of a VQ-VAE is defined as $\mathcal{Z} := \{1, 2, \ldots, K\}^{H \times W}$, yielding a latent space of an exponential size, $|\mathcal{Z}| := K^{HW}$.*

The VQ-VAE's prior $p(\mathbf{z})$ is learned via an autoregressive model, such as PixelCNN [van den Oord et al., 2016a,b]. Viewing the discrete latent variable as a sequence of indices, the prior is modeled as $p(\mathbf{z}|c) := \prod_{i=1}^{HW} p(z_i|\mathbf{z}_{<i}, c)$, where $\mathbf{z}_{<i}$ are all indices before $i$ in the row-major order, and $c$ is a high-level data description represented as a latent vector (e.g., a class label in supervised learning). VQ-VAEs are an example of the discrete latent variable model in (1), where the PixelCNN prior parametrizes $p(\mathbf{z}) := \sum_c p(c)p(\mathbf{z}|c)$, and the decoder defines $p(\mathbf{x}|\mathbf{z})$ using a deep neural network. For further details on VQ-VAE, refer to Appendix A.

## 3 DISTILLING MIXTURE MODELS

**VQ-VAE intractability.** Computing $p(\mathbf{x}|\mathbf{z})$ for all $\mathbf{z} \in \mathcal{Z}$ is very expensive since $p(\mathbf{x}|\mathbf{z})$ is typically parametrized by

a large neural network and the size of the latent space is exponentially high (Definition 1). Consequently, the training of the VQ-VAE via the exact likelihood (1) and the ELBO [Kingma and Welling, 2022] objectives is infeasible. To deal with this problem, the training is done by a heuristic loss function [van den Oord et al., 2017], see Appendix A. This also implies that computing any probabilistic inference queries with (1) is intractable.

**Model distillation.** We propose to address this intractability issue by distilling the VQ-VAE model into a mixture of tractable distributions. We refer to this model as a distilled model (DM). The main motivation behind this idea is that index collapse results in a substantial underutilization of the latent space, as shown in Figure 1. Therefore, we create a subset of most representative latent variables $\bar{\mathcal{Z}} \subset \mathcal{Z}$ and construct the DM as follows:

$$\hat{p}(\mathbf{x}) := \sum_{\mathbf{z} \in \bar{\mathcal{Z}}} \frac{1}{|\bar{\mathcal{Z}}|} p(\mathbf{x}|\mathbf{z}). \tag{2}$$

We discuss two ways to construct $\bar{\mathcal{Z}}$, but, first, we state assumptions that ensure tractability of the DM.

**Assumption 1.** $\mathbf{x}$ *is conditionally independent given* $\mathbf{z}$*, i.e.,* $p(\mathbf{x}|\mathbf{z}) := \prod_{i=1}^{S} p(x_i|\mathbf{z})$*, and each* $p(x_i|\mathbf{z})$ *is a tractable distributions (e.g., Gaussian, categorical).*

Under the conditional-independence (Assumption 1), the discrete latent variable model (1) reveals the connection between VQ-VAEs and PCs. Indeed, there can be many equivalent representation between these two models, depending on a specific architecture of a PC. However, the simplest one is that the decoder $p(\mathbf{x}|\mathbf{z})$ can be seen as a product unit whose children are input units, and that the PixelCNN prior $p(\mathbf{z})$ represents the weights of a sum unit. To make the resulting DM tractable, we also have to satisfy the following assumption.

**Assumption 2.** *The number of components in* (2) *is kept computationally feasible, i.e.,* $N \ll K^{HW}$*.* [1]

**Random sampling.** One way to construct the latent set $\bar{\mathcal{Z}}$ is to enumerate $p(\mathbf{z})$ for all $\mathbf{z} \in \mathcal{Z}$, and then sample a distinct set of latent variables from the class-conditioned VQ-VAE prior, $\bar{\mathcal{Z}} = \{\mathbf{z}_i | c_i \sim \mathcal{U}(\mathcal{C}), \mathbf{z}_i \sim p(\mathbf{z}|c_i)\}_{i=1}^{N}$. The DM built from randomly sampled latent variables is called a DM via Random Sampling (DMRS). However, the exhaustive enumeration is impractical due to the large latent space (Definition 1).

**Beam search.** To overcome the drawbacks of DMRS, $\bar{\mathcal{Z}}$ can be constructed by a guided search through the latent space. This traversal of $\mathcal{Z}$ identifies the most probable, informative regions without the exhaustive enumeration. Viewing

---

[1] The original model in (1) is recovered for $N = K^{HW}$.

the $H \times W$ latent grid as a row-major sequence of $HW$ tokens over a vocabulary of size $K$, allows the application of sequence search techniques, such as beam search. Beam search (BS) is commonly employed to maintain tractability in large search spaces by trading off completeness and optimality [Xu et al., 2009]. The latent space of VQ-VAE (Definition 1) is one such example where BS can be applied due to its size. The class-conditioned stochastic BS [Shao et al., 2017], (Algorithm 1), discovers latent variables for distillation, resulting in the DM via Beam Search (DMBS). DMBS requires $\mathcal{O}(NHWK)$ operations, i.e., dramatically fewer than $\mathcal{O}(K^{HW})$ needed for the exhaustive enumeration. This trade-off manifests in reduced expressivity, but, as we demonstrate in (Section 4), its effects are modest.

---

**Algorithm 1** Class-Conditioned Beam Search

See the BeamSearch implementation in Appendix D for details.
**Input:** $N$ sequences, $s$ initial samples, class set $\mathcal{C}$
**Output:** Candidate set $\bar{\mathcal{Z}}$
$\bar{\mathcal{Z}} \leftarrow \emptyset$
**for** $c \in \mathcal{C}$ **do**
$\quad$ **for** $i \leftarrow 1$ **to** $s$ **do**
$\quad\quad \mathbf{z} = \{0\}^{H \times W}$
$\quad\quad z \sim p(\mathbf{z}_1 \mid c)$
$\quad\quad \mathbf{z}_1 = z$
$\quad\quad \bar{\mathcal{Z}} \leftarrow \bar{\mathcal{Z}} \cup \text{BeamSearch}\left(\mathbf{z}, p, \{c\}, H, W, B \leftarrow \frac{N}{s|\mathcal{C}|}, i_1 \leftarrow 2\right)$
**return** $\bar{\mathcal{Z}}$

---

# 4 EXPERIMENTAL RESULTS

We demonstrate the tractability of our DMs on two core probabilistic-inference tasks: density estimation and image inpainting. The goal is to exhibit that the DMs can, quickly and accurately, answer complex probabilistic queries, yielding answers that approach state-of-the-art models. Evaluations are conducted on MNIST [Lecun et al., 1998], modeling the pixels by the Gaussian distribution. We refer the reader to Appendix B for details about data pre-processing.

**Models.** We choose two tractable probabilistic models as baselines: continuous mixtures (CMs)[Correia et al., 2023] and Einsum networks (EiNets)[Peharz et al., 2020]. CMs approximate $p(\mathbf{x})$ as an uncountable mixture of $p(\mathbf{x}|\mathbf{z})$ components integrated over a continuous latent variable, $\mathbf{z} \in \mathbb{R}^D$. Approximating the integral with a finite set of points compiles the model into a PC, aligning its structure with that of the DMs. EiNets are tensorised PCs optimised for parallel computing on contemporary GPUs, facilitating a simple design of deep and tractable generative models. CMs and EiNets represent cutting-edge baselines in density estimation and image inpainting, demonstrating strong performance across benchmarks [Correia et al., 2023, Peharz et al., 2020]. The exact model (ExM), which exhaustively enumerates the latent space $\mathcal{Z}$ to compute the exact likelihood in (1), serves as a principled performance baseline for evaluating the DMs, instantiations of (2), and highlights the performance gap relative to exact evaluations of $p(\mathbf{x})$.

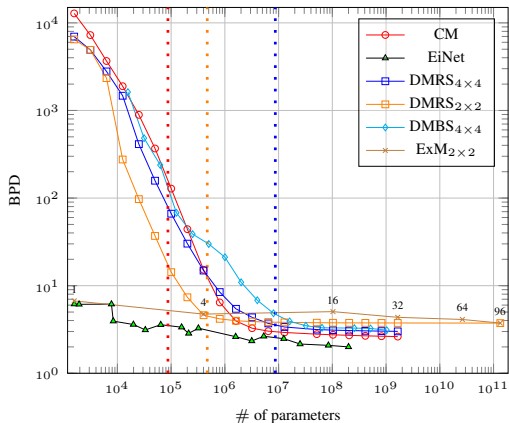

Figure 2: *BPD performance vs model size for different tractable models.* Lower is better. Distilled model is denoted with hollow squares, while EiNets have filled markers, as they are trained PCs. Dotted vertical lines indicate the sizes of the source VQ-VAEs for the distilled model. For the continuous mixtures, it shows the decoder size. The number labels express the VQ-VAE codebook sizes, $K$, for the exact models. All results are averaged over 5 runs with different random seeds.

**Settings.** The DMs, $DM_{2\times2}$, $DM_{4\times4}$, and $DM_{7\times7}$, use latent grids of $1\times2\times2$ with 96 codewords, $128\times4\times4$ with 512 codewords, and $128\times7\times7$ with 1024 codewords, respectively. The exact models, $ExM_{2\times2}$, share a latent size $1\times2\times2$ varying only in codebook size $K \in \{1, 4, 16, 32, 64, 96\}$. With up to $96^4$ components, this is the largest configuration for which exact $p(\mathbf{x})$ is still feasible given the exponential latent space growth (see Definition 1); beyond this, full enumeration is impractical despite further MNIST performance gains. The DMBS model sampled $s = 10$ starting $\mathbf{z}_{1,1}$ values. The CM latent space of dimension is $16\times1\times1$, with $2^{14}$ integration points, used in Correia et al. [2023]. The EiNets adopt a single-layer design, using the PoonDomingos structure [Poon and Domingos, 2012] for decomposition. We refer the reader to Appendix C for details about the training.

**Density estimation.** Figure 2 compares all the aforementioned models in terms of bits per dimension (BPD). We can see that the $ExM_{2\times2}$ model sets an exact performance bound which is quickly approached by the $DMRS_{2\times2}$ model, indicating that even the randomized distillation captures critical information encoded by the VQ-VAE. Importantly, we can see that the $DMBS_{4\times4}$ model closely follows the $DMRS_{4\times4}$ model, which demonstrates that the beam search has the ability to select important parts of the latent space without the exhaustive enumeration necessary for the random sampling. The CM model outperforms all the DM models; however, as shown in Appendix E (and also Figure 3), its sample quality is lower (see Appendix C for architecture details). We conjecture that this is attributed to the smaller size of the CM's decoder (the red dashed line), from which the model is compiled. The performance of all the DMs plateaus, which shows that distilling more of the same or similar latent variables does not bring additional information into the resulting DMs. Interestingly, the EiNet model

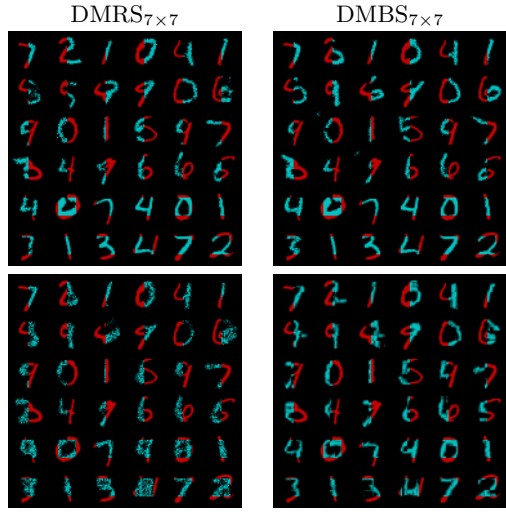

Figure 3: *Image inpainting by tractable probabilistic models.* The unobserved $\mathbf{x}_u$ and observed $\mathbf{x}_o$ parts are highlighted by the red and blue colors, respectively.

outperforms all the other models for all parameter counts; however, its sample quality seems lower (Figure 3). We offer additional experiments in Appendix F.

**Tractable inference.** We demonstrate the tractability of the DMs through the image inpainting, which corresponds to the conditional inference task $p(\mathbf{x}_u|\mathbf{x}_o)$, where $\mathbf{x}_u$ and $\mathbf{x}_o$ are unobserved and observed image parts, respectively. Unlike VQ-VAEs solely specialized for inpainting [Peng et al., 2021], the DM supports diverse inference tasks, including marginalization, expectation, and maximum a posteriori estimation. Figure 3 compares inpainting of MNIST images for different models. It can be seen that all models successfully infill missing parts to form correct digits. Importantly, the image quality of the reconstructions done by DMs is mostly better.

## 5 CONCLUSIONS

We have proposed a novel framework for distilling tractable mixtures from otherwise intractable VQ-VAEs. Our model is able to answer a broad range of probabilistic inference tasks—the same as with the conventional PCs—while closely retaining the expressive power of VQ-VAEs. We have investigated two strategies for identifying informative latent space regions: the random sampling and the beam search. Though the random sampling has proven efficient, its key disadvantage is the full enumeration of the latent space, making it impractical for scaling to state-of-the-art VQ-VAEs. Importantly, the beam search delivers almost identical performance, while avoiding this exhaustive enumeration. In future work, we plan to design data-driven exploration strategies of the latent space that will allow us to further improve the performance of our distilled model.

# ACKNOWLEDGEMENTS

The authors acknowledge the support of the National Recovery Plan funded project MPO 60273/24/21300/21000 CEDMO 2.0 NPO and the OP VVV funded project CZ.02.1.01/0.0/0.0/16_019/0000765 "Research Center for Informatics".

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

# Distillation of a tractable model from the VQ-VAE
## (Supplementary Material)

**Armin Hadžić** [1]               **Milan Papež**[1]               **Tomáš Pevný**[1]

[1]Artificial Intelligence Center, Czech Technical University. Prague, Czech Republic.

## A   VQ-VAE

For completeness, an extended description of the VQ-VAE, introduced in Section 2, is provided, including its encoder E, decoder D and vector quantizer block VQ. The encoder is a deep neural network $E : \mathbb{R}^{d \times h \times w} \to \mathbb{R}^{D \times H \times W}$, compresses $\mathbf{x}$ into a continuous latent variable $E(\mathbf{x})$. Here, $H$ and $W$ define the spatial dimensions of the latent grid and $D$ is the size of each codeword. The vector quantizer then discretizes the continuous to a discrete latent variable $\mathbf{z} \in \mathcal{Z}$ by performing a nearestneighbor search in the learned codebook $\{\mathbf{e}_i\}_{i=1}^K \subset \mathbb{R}^D$. The vector quantizer assigns each latent position $(i, j)$ to the nearest codebook embedding $\mathbf{e}_k$ according to Euclidean distance, $VQ : \mathbb{R}^{D \times H \times W} \to [K]^{H \times W}$, where $[K] := \{1, 2, \ldots, K\}$. This induces a categorical posterior distribution with one-hot probabilities over the discrete latent variables, where for each $(i, j) \in [H] \times [W]$ the posterior is given by

$$q\big(\mathbf{z}_{i,j} = k \mid \mathbf{x}\big) = \begin{cases} 1, & \text{if } k = \underset{l \in [K]}{\operatorname{argmin}} ||E(\mathbf{x})_{:,i,j} - \mathbf{e}_l||_2, \\ 0, & \text{otherwise.} \end{cases} \tag{3}$$

This quantization process can be divided into the following steps: (i) pass $\mathbf{x}$ through the encoder network to obtain the prequantized latent representation $E(\mathbf{x})$, (ii) compare $E(\mathbf{x})$ to all latent embeddings $\mathbf{e}_j$ in the codebook, and (iii) assign $\mathbf{z}_{i,j} = k$ as the embedding index $\mathbf{e}_k$ with the smallest Euclidean distance to $E(\mathbf{x})_{:,i,j}$. The decoder $D : \mathbb{R}^{D \times H \times W} \to \mathbb{R}^{d \times h \times w}$ is a deep neural network that reconstructs the input $\mathbf{x}$, producing $\hat{\mathbf{x}}$, by decompressing the discrete latent variable $\mathbf{z}$. This process is split into the following two steps: (i) mapping of each discrete index in $\mathbf{z}$ into its corresponding embedding codeword $\mathbf{e}_k$ forming the post-quantized continuous latent variable, and (ii) decoding the post-quantized continuous latent variable to produce the reconstruction $\hat{\mathbf{x}}$. The decoder distribution is given by

$$p(\mathbf{x}|\mathbf{z} = \mathbf{e}_k), \quad \text{where } k = \underset{j \in [K]}{\operatorname{argmin}} ||E(\mathbf{x}) - \mathbf{e}_j||_2, \tag{4}$$

and models the conditional likelihood of the observed data given the latent code. Decoder parameterizes $p(\mathbf{x}|\mathbf{z})$ and its specific form depends on the modality of $\mathbf{x}$, such as Bernoulli for binary data or Gaussian distribution for continuous-valued observations. The Variational Autoencoder, which the VQ-VAE parallels, is typically trained by maximising the Evidence Lower Bound objective

$$\log p(\mathbf{x}) \geq \mathbb{E}_{q(z|\mathbf{x})}[\log p(\mathbf{x}|\mathbf{z})] - D_{KL}(q(\mathbf{z}|\mathbf{x})||p(\mathbf{z})), \tag{5}$$

which is a tractable surrogate to the intractable exact likelihood. However, due to the combinatorial complexity of the latent space (see Definition 1), the KL divergence term in (5) renders the objective intractable for the VQ-VAE model. This problem is circumvented in van den Oord et al. [2017] by introduction of the following tractable heuristic loss function

$$\mathcal{L}(\mathbf{x}) = \underbrace{||\mathbf{x} - \hat{\mathbf{x}}||_2^2}_{\text{reconstruction}} + \underbrace{||\text{sg}[E(\mathbf{x})] - \mathbf{e}||_2^2}_{\text{codebook loss}} + \beta \underbrace{||E(\mathbf{x}) - \text{sg}[\mathbf{e}]||_2^2}_{\text{commitment loss}}. \tag{6}$$

This objective comprises three components: (i) a reconstruction loss, which measures how well the decoder reconstructs $\mathbf{x}$ from $\mathbf{z}$, (ii) a codebook loss, encouraging codebook embeddings $\mathbf{e}$ to move toward the encoder output (codebook

*Accepted for the 8$^{th}$ Workshop on Tractable Probabilistic Modeling at UAI* (TPM 2025).

learning), and (iii) a commitment loss, which forces the encoder outputs to commit to specific codebook entries. Here, $\text{sg}[\cdot]$ denotes the stop-gradient operator, which blocks gradients during backpropagation to ensure stability in training and $\beta$ is a hyperparameter balancing the commitment loss.

## B    DATASET

The MNIST dataset consists of grayscale images with discrete pixel intensities $\mathbf{x} \in \{0, 1, \ldots, 255\}^{28 \times 28}$. Following prior work Theis et al. [2016], Correia et al. [2023], we apply uniform jittering and scaling to map pixel values to the continuous range $\mathcal{X} := [0, 1]^{28 \times 28}$. We evaluate generative models using the bits per dimension (BPD) metric, which quantifies the average number of bits required to encode each pixel in an image. Because the pixel values are rescaled to $[0, 1]$, we adjust the BPD computation accordingly:

$$\text{BPD}(\{\mathbf{x}^{(i)}\}_{i=1}^n) = \frac{1}{n} \sum_{i=1}^{n} \left( -\frac{\log_2 p\left(\mathbf{x}^{(i)}\right)}{S} + 8 \right), \tag{7}$$

where $S = 784$ is the total number of pixels in $\mathbf{x}$.

## C    MODEL AND TRAINING SETTINGS

The EiNets use a single-layer PoonDomingos image decomposition structure. We use PoonDomingos piece sizes $\{\{2\}, \{4\}, \{7\}\}$, with the number of input distributions per patch set to $I \in 1, 5, 10, 20, 30, 40$.

VQ-VAE and PixelCNN models were implemented in PyTorch [Paszke et al., 2019] and trained for up to 100 epochs, with the Adam optimizer [Kingma and Ba, 2015], employing early stopping on validation loss with a maximum of 15 epochs.

The VQ-VAE models use a convolutional encoder-decoder architecture. The encoder consists of convolutional layers with stride for downsampling, followed by batch normalization and ReLU activations. Residual blocks are used at the latent level. The decoder mirrors the encoder, using residual blocks followed by transposed convolutions for upsampling, again with batch normalization and ReLU. A final activation is applied based on the data modality: for Gaussian modeling, two output channels represent the mean (with sigmoid activation) and log-variance (clamped to ensure standard deviation lies in $[10^{-3}, 1]$); for discrete pixels, a softmax outputs class probabilities. The encoder-decoder architecture used in our VQ-VAE models differs from the one used in CMs [Correia et al., 2023], as the latent space used by CMs is not sufficient to effectively fit a VQ-VAE. VQ-VAEs benefit from using larger spatial latent shapes, such as $4 \times 4$ or $7 \times 7$, compared to $1 \times 1$ used by CMs, to compensate for the limitations introduced by the discrete latent space and vector quantization.

## D    BEAM SEARCH

Algorithm 2 outlines a beam search procedure for generating high-probability sequences of discrete latent variables $\mathbf{z}$, as defined by the probabilistic model $p(\mathbf{z})$. The search maintains a beam, a collection of the top-$B$ candidate sequences, ranked by their marginal likelihood scores over the class set $\mathcal{C}$. A key feature of this beam search implementation is its flexibility: it can run as a standard beam search from the beginning of the sequence or resume from any intermediate position. In the latter case, the initial segment of the sequence must be pre-initialized, and the search continues from the index $i_1$, completing the remaining latent grid. This enables conditional sampling or constrained generation, where parts of the latent representation are fixed, e.g., from observed data or previous decisions, and the algorithm completes the rest in a manner consistent with the beam search and likelihood scores given by $p$. Each candidate sequence in the beam is extended, element by element, in row-major order across the $H \times W$ grid, scored, and pruned to retain only the top $B$ sequences at each step. The final beam $\mathcal{B}_{HW}$ contains the top-scoring complete sequences, which form the set $\bar{\mathcal{Z}}$

**Algorithm 2** Beam Search

**Input:** Discrete latent variable $\mathbf{z}$; Prior model $p(\mathbf{z})$; Set of target classes $\mathcal{C}$; Latent grid dimensions $H \times W$; Beam width $B$; Starting position index $i_1$
**Output:** Set of latent variables $\tilde{\mathcal{Z}}$
$h_c = p\big(\mathbf{z}_{i_1} = k \mid \mathbf{z}_{<i_1}, c\big), \forall c \in \mathcal{C}$
$m = 0$
$\mathcal{B}_1 = \big\{ (\mathbf{z}, h, m) \big\}$
**for** $i = i_1, \ldots, HW$ **do**
    $\widehat{\mathcal{B}}_i \leftarrow \emptyset$
    **foreach** $(\mathbf{z}, h, m) \in \mathcal{B}_{i-1}$ **do**
        **for** $k = 1, \ldots, K$ **do**
            $h'_c = h_c p\big(\mathbf{z}'_i = k \mid \mathbf{z}'_{<i}, c\big), \quad \forall c \in \mathcal{C}$
            $m' = \sum_{c \in \mathcal{C}} p(c) h'_c$
            $\mathbf{z}'_i = k$
            $\widehat{\mathcal{B}}_i \leftarrow \widehat{\mathcal{B}}_i \cup \{(\mathbf{z}', h', m')\}$
    $\mathcal{B}_i = \big\{(\mathbf{z}^{(t)}, h^{(t)}, m^{(t)})\big\}_{t=1}^{B}, \text{s.t.} m^{(1)} \geq m^{(2)} \geq \cdots \geq m^{(|\widehat{\mathcal{B}}_i|)}$
**return** $\big\{(\mathbf{z}^{(t)})\big\}_{t=1}^{B}$    s.t.   $m^{(1)} \geq m^{(2)} \geq \cdots \geq m^{(|\mathcal{B}_{HW}|)}$

# E  SAMPLE QUALITY

Figure 4 presents example images generated by the four evaluated tractable models. All models are capable of producing digit-like images to varying degrees. EiNets yield the highest-quality samples, while the randomly sampled DMRS model also generates visually plausible digits. CMs perform the worst in terms of sample fidelity, while the DM models, both DMBS and DMRS provide decent samples. The quality of generated data was evaluated extrinsically using a simple

$\text{DMRS}_{7\times7}$          $\text{DMBS}_{7\times7}$

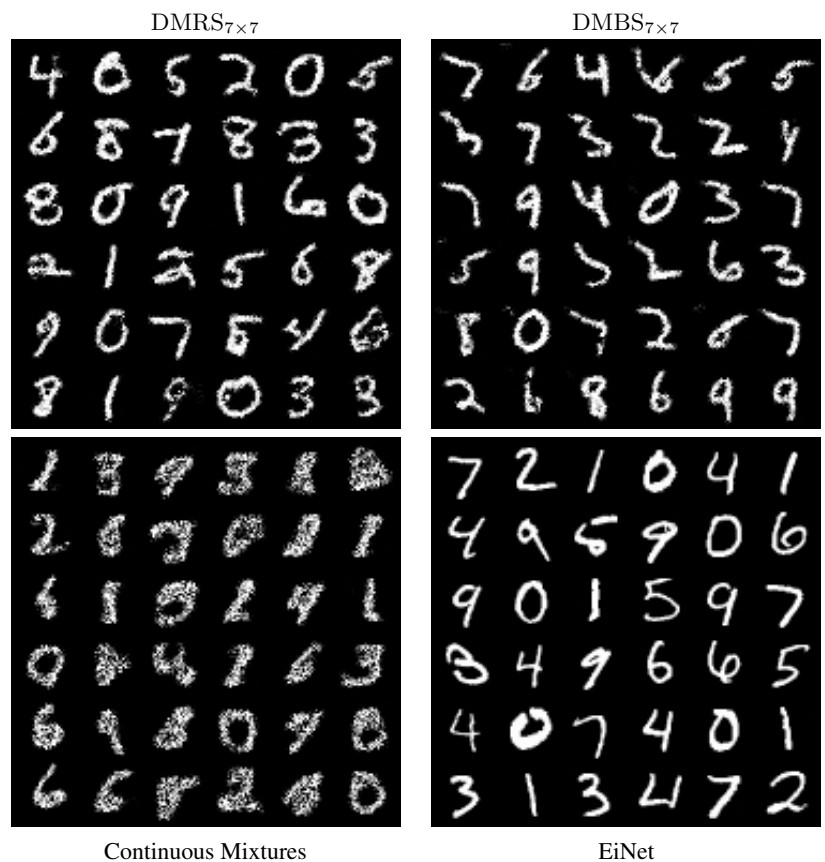

Continuous Mixtures          EiNet

Figure 4: *Image sample by tractable probabilistic models.*

MNIST classifier. Each model produced $10\,000$ samples, which were subsequently classified to assess how closely the resulting class distributions aligned with the classifiers predictions on the MNIST dataset. Since the true MNIST distribution is approximately uniform, an ideal generative model should produce a similarly uniform class distribution, otherwise it indicates systematic biases in the generation process. The classification results are summarized in Figure 5, which includes

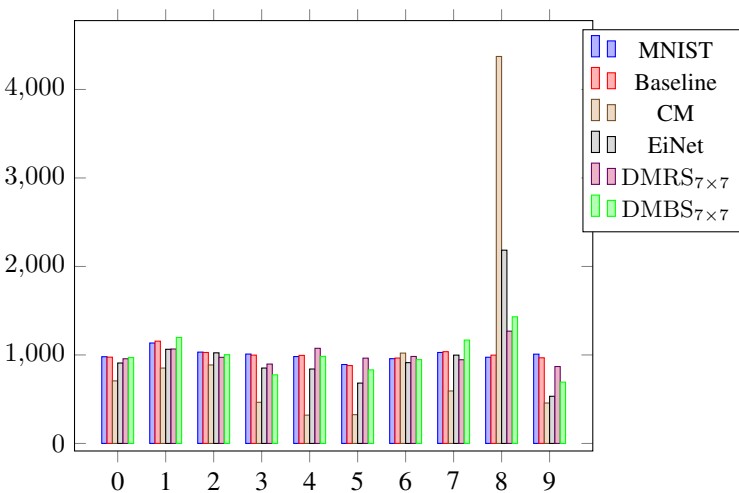

Figure 5: *Histogram of label distributions of generated MNIST samples.* Generated samples were classified by a pre-trained MNIST classifier, with bars grouped by class label. The blue bars represent the MNIST dataset distribution, while the red bars show the baseline classifier performance on MNIST data, demonstrating close alignment that validates the classifier's reliability. Subsequent bars display results for generated samples: brown corresponds to Continuous Mixture (CM), gray to EiNet, purple to DMRS, green to DMBS. The x-axis indicates MNIST digit classes and the y-axis shows sample counts in log scale.

the baseline class distribution from the MNIST dataset for reference. Interestingly, all generative models exhibit comparable trends, with the exception being DMBS model. Notably, the DMRS model achieves the closest match to the uniform baseline, indicating high-quality and balanced sample generation. In contrast, the CM model exhibits a noticeable deviation, with a pronounced bias toward generating the digit eight at the expense of other digits, as illustrated in Figure 4.

## F  LATENT CORRELATION

Distilling a mixture model from the set of candidate latent variables $\bar{\mathcal{Z}}$ by selecting latent variables solely based on their highest $p(\mathbf{z})$ values results in highly correlated latent variables. High correlation among mixture components impairs density estimation, as it tends to favor modes that capture dominant patterns, reinforcing similarity among components. Ideally, mixture components should specialize in different regions of the data space. To analyze the correlation of latent likelihoods across components, we consider a setup with 4 mixture components. For each component, we compute the joint likelihoods $p(\mathbf{x}, \mathbf{z})$ over 500 test samples, resulting in a total of $4 \times 500$ likelihood values. We visualize these values using corner plots, in Figure 6, to examine how likelihoods are distributed across components and whether strong correlations emerge. This analysis helps assess the diversity and independence of the mixture components. The left plots reveal distinct modes and diverse density regions, indicating that components specialize in different parts of the input space and contribute more evenly to the mixture. In contrast, the right plots exhibit limited dispersion and overlap across components, reflecting reduced diversity and weaker mixture modeling capacity.

## G  CATEGORICAL DATA

This appendix presents additional results of DMs applied to categorical MNIST, where pixel intensities are modeled as discrete values. We evaluate the tractable modelsCMs, EiNets, and DMson two tasks: density estimation and image inpainting. Figure 7 shows the bits-per-dimension (BPD) performance across models, while Figure 8 illustrates qualitative inpainting examples, highlighting the capacity of each model to infer missing image regions from partial observations.

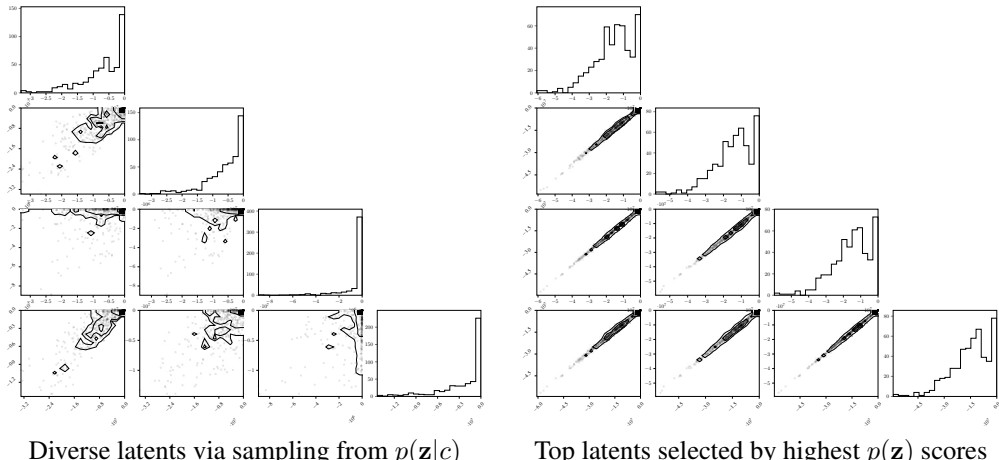

Diverse latents via sampling from $p(\mathbf{z}|c)$      Top latents selected by highest $p(\mathbf{z})$ scores

Figure 6: *Corner plots illustrating latent correlations in DM models.* Left: Corner plots of the $\mathrm{DMRS}_{2\times2}$ model, where latent samples $\mathbf{z} \sim p(\mathbf{z}|c)$ are drawn randomly. The contour shapes here are less aligned and more scattered. This suggests that the selected components respond more independently to the data, indicating lower correlation among latents. Right: Corner plots of the $\mathrm{DM}_{2\times2}$ model, where the top 4 latent codes are selected based on the highest values of $p(\mathbf{z})$. The elongated contours along the diagonals in each subplot suggest that the log-likelihoods of different components are highly correlated. This indicates redundancy in the selected components, meaning they often respond similarly to the same inputs. Both models share the same VQ-VAE architecture and PixelCNN prior model.

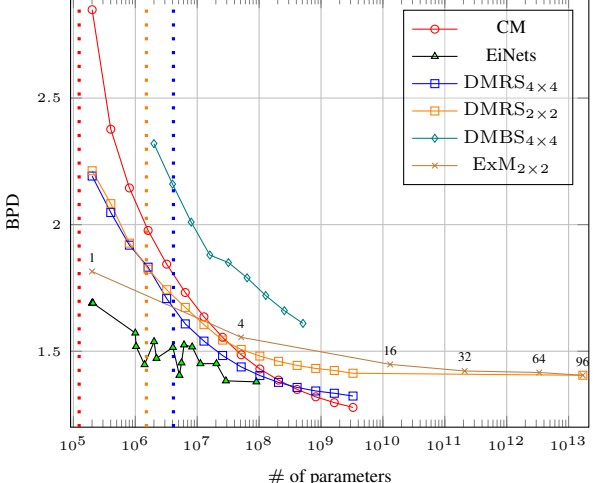

Figure 7: *BPD performance vs model size for different tractable models for categorical data.* Lower is better. All distilled models are denoted with hollow, while EiNets have filled markers, as they are trained rather than distilled PCs. Dotted vertical lines indicate the sizes of the source VQ-VAEs for distilled models. For the continuous mixtures, it shows the decoder size. The number labels express the VQ-VAE codebook sizes, $K$, for the exact models.

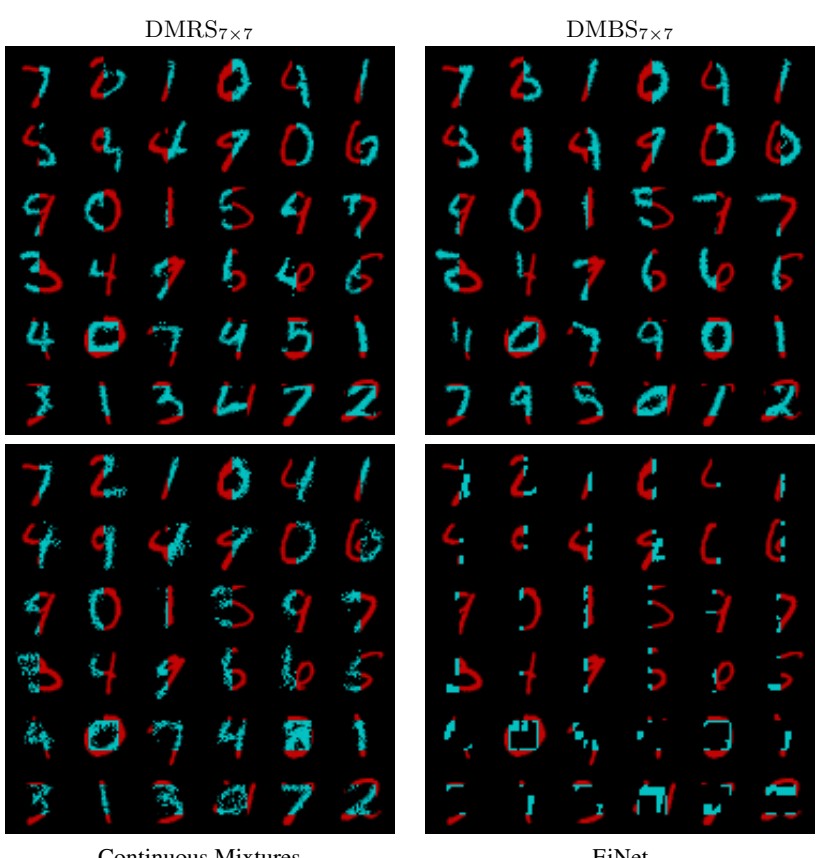

Figure 8: *Image inpainting by tractable probabilistic models for categorical data.* The unobserved $\mathbf{x}_{\mathrm{u}}$ and observed $\mathbf{x}_{\mathrm{o}}$ parts are highlighted by the red and blue colors, respectively.