# OpenReview forum: "Distillation of a tractable model from the VQ-VAE"
_auai.org/UAI/2025/Workshop/TPM — TPM 2025_

### Official Review · Reviewer_uQMa · 2025-06-08
**Distilling PCs from VQ-VAEs**

**Rating:** 3

**Review:**

I enjoyed reading the paper, which is quite relevant for the workshop.

Some commets/questions:
- The major drawback of the method is that requires having access to a prior p(z), like a PixelCNN/transformer, which is something that is usually trained after the VQ-VAE. Moreover, such a prior is *not* usually conditioned on class information, while the BeamSearch method seems to require it, see Algorithm 2. I also did not find information about the prior used in the experiments. Can the authors comment on this? is it a conditional PixelCNN?
- The original continuous mixture (CM) method is fully unsupervised. Did the authors change the architecture of the decoder to take into account class information? If so, how? If not, then I think comparisons might be unfair, as the PCs are distilled using a conditional prior that relies on class information
- Can the authors report BPD values in tabular form?
- the MNIST samples from the EiNet model (appendix) look really great. Is then EiNet trained using class information (and so discriminatively)? Also, I'm wondering what is a single-layer Poon–Domingos image decomposition structure? can the authors provide more details about this architecture?

---

### Official Review · Reviewer_3oyv · 2025-06-11

**Rating:** 3

**Review:**

The paper proposes to select a subset of the codebook of a VQ-VAE to enable tractable inference via a probabilistic circuit. The authors propose two methods to search codebook for the most relevant latent vectors with random or beam search. The distilled PC shows good sample quality and density estimation in experiments with continuous and categorical data.

### Strengths

The idea of using the probabilistic circuits to do exact inference on an approximate VQ-VAE is quite elegant and seems to work well in practice both in terms of sample quality and density estimation. The paper is well written and the experiments well designed.

### Weaknesses

The only important weakness of the paper is that experimental results are limited to MNIST, which is a very simple dataset. It would be nice to see larger and more complex datasets for future versions of the paper. Comparisons to other methods, like PICs (Gala et al. 2024) could also be interesting.

### Questions

1. Am I correct to assume the method requires no training? The decoder is the same as from VQ-VAE, and it suffices to select the relevant subset of the codebook?
2. Even though the latent space of VQ-VAE is known to be underutilized, would it make it sense to include a regularization during training such that the number of latent codes that are effectively used match the PC budget? Could that improve performance?
3. Comparing Figure 2 and 7, we see that DMBS performs much better on continuous data than on categorical data. Why could that be?
4. The experiment in Figure 5 is quite interesting. Any thoughts on what induces the bias in continuous mixtures and why DMRS does not suffer the same effect?
5. The performance of Einets seems better than expected, especially with a simple Poon-Domingos architecture. Do the authors have any thoughts as to why the proposed method and other baselines seem to underperform Einets in terms of BPD and sample quality?

### References

Gala, Gennaro, et al. "Probabilistic integral circuits." International Conference on Artificial Intelligence and Statistics. PMLR, 2024.